# HER2-Low Breast Cancer at the Interface of Pathology and Technology: Toward Precision Management

**DOI:** 10.3390/biomedicines14010049

**Published:** 2025-12-25

**Authors:** Faezeh Shekari, Reza Bayat Mokhtari, Razieh Salahandish, Manpreet Sambi, Roshanak Tarrahi, Mahsa Salehi, Neda Ashayeri, Paige Eversole, Myron R. Szewczuk, Sayan Chakraborty, Narges Baluch

**Affiliations:** 1Department of Biomedical and Molecular Sciences, Queen’s University, Kingston, ON K7L 3N6, Canada; faezehshekari@gmail.com (F.S.); roshanak.tarrahi@gmail.com (R.T.); s.mahsasalehi@gmail.com (M.S.); neda.ashayer@gmail.com (N.A.); szewczuk@queensu.ca (M.R.S.); 2Department of Pharmacology and Therapeutics, Roswell Park Comprehensive Cancer Center, Buffalo, NY 14263, USA; paige.eversole@roswellpark.org (P.E.); syan.chakraborty@roswellpark.org (S.C.); 3Association of Clinical Immunology and Cancer Research (ACICR), San Diego, CA 92127, USA; 4California Comprehensive Allergy and Food Institute, P.C. (CalCafi, P.C.), San Diego, CA 92131, USA; 5Laboratory of Advanced Biotechnologies for Health Assessments (Lab-HA), Biomedical Engineering Program, Lassonde School of Engineering, York University, Toronto, ON M3J 1P3, Canada; raziehs@yorku.ca; 6Department of Electrical Engineering and Computer Science (EECS), Lassonde School of Engineering, York University, Toronto, ON M3J 1P3, Canada; 7Department of Immunology and Allergy, Rady Children’s Hospital, San Diego, CA 92123, USA

**Keywords:** HER2-low breast cancer, liquid biopsy, clinicopathologic traits, hormone receptor status, precision diagnostics, biosensors

## Abstract

**Background/Objectives:** HER2-low breast cancer has emerged as a clinically meaningful category that challenges the historical HER2-positive versus HER2-negative classification. Although not defined as a distinct biological subtype, HER2-low tumors exhibit unique clinicopathological features and differential sensitivity to novel antibody–drug conjugates. Accurate identification remains difficult due to limitations in immunohistochemistry performance, inter-observer variability, intratumoral heterogeneity, and dynamic shifts in HER2 expression over time. This review synthesizes current evidence on the biological and clinical characteristics of HER2-low breast cancer and evaluates emerging diagnostic innovations, with emphasis on liquid biopsy approaches and evolving technologies that may enhance diagnostic accuracy and monitoring. **Methods**: A narrative literature review was conducted, examining tissue-based HER2 testing, liquid biopsy modalities, including circulating tumor cells, circulating nucleic acids, extracellular vesicles, and soluble HER2 extracellular domains, and applications of artificial intelligence (AI) across histopathology and multimodal diagnostic systems. **Results**: Liquid biopsy technologies offer minimally invasive, real-time assessment of HER2 dynamics and may overcome fundamental limitations of tissue-based assays. However, these platforms require rigorous analytical validation and face regulatory and standardization challenges before widespread clinical adoption. Concurrently, AI-enhanced histopathology and multimodal diagnostic systems improve reproducibility, refine HER2 classification, and enable more accurate prediction of treatment response. Emerging biosensor- and AI-enabled monitoring frameworks further support continuous disease evaluation. **Conclusions**: HER2-low breast cancer sits at the intersection of evolving pathology and technological innovation. Integrating liquid biopsy platforms with AI-driven diagnostics has the potential to advance precision stratification and guide personalized therapeutic strategies for this expanding patient subgroup.

## 1. Introduction

Breast cancer remains a significant global challenge, with ongoing efforts to improve diagnosis, treatment, and patient outcomes through a deeper understanding of its underlying mechanisms. The identification of human epidermal growth factor receptor 2 (HER2) as a crucial prognostic and therapeutic marker in breast cancer has revolutionized clinical practice by enabling the development of targeted therapies. Although HER2 is also known as erythroblastic oncogene B-2 (ERBB2), Neu, and CD340, we primarily refer to it as HER2 in this review.

Traditionally, HER2 status in breast cancer has been evaluated dichotomously as either positive or negative to determine eligibility for anti-HER2 therapies [1]. A significant development in the field is the recognition that roughly 50–60% of breast tumors previously classified as HER2-negative actually express low levels of HER2, which is now referred to as HER2-low breast cancer [1,2]. This discovery has led to a paradigm shift in how we conceptualize and treat breast cancer, opening up new avenues for targeted therapies [2,3,4,5,6,7].

The evolving landscape of HER2-low breast cancer has transformed it from a category with no specific treatment options to one with potential molecular targets for therapy, as extensively discussed in previous reviews [8,9,10,11]. As our understanding of breast cancer subtypes continues to grow, there is potential for more targeted therapies and personalized treatment options for patients with HER2-low breast cancer. Recent studies have further refined our understanding of HER2-low breast cancer, suggesting that it may represent a distinct biological entity with unique clinical implications [12,13].

In this review, we examine HER2-low breast cancer “at the interface of pathology and technology.” We first trace the transition from classic HER2-positive/negative classification to the evolving HER2-low subtype, explore its relationship with hormone receptor status, and summarize current and emerging therapeutic strategies. We then address the practical challenges of HER2 detection, highlight the expanding role of liquid biopsy and its next-wave innovations, and discuss technology-driven solutions to critical clinical questions. By integrating these perspectives, we outline a roadmap toward precision management of this increasingly recognized breast cancer subtype.

## 2. Transitioning from Classic HER2-Positive/Negative Classification to the Evolving HER2-Low Subtype

HER2 status is traditionally evaluated using immunohistochemistry (IHC) scoring: 3+ (complete, intense membrane staining in >10% tumor cells), 2+ (complete membrane staining, weak to moderate intensity), 1+ (incomplete, faint membrane staining), and 0 (no detectable staining). Per 2018 the American Society of Clinical Oncology (ASCO)/the College of American Pathologists (CAP) guidelines, HER2-positive tumors (IHC 3+ or in-situ hybridization (ISH)-amplified) receive targeted therapy, while HER2-negative cases (IHC 0/1+ or 2+ ISH-negative) traditionally do not. However, revolutionary artificial intelligence (AI) solutions have emerged as the definitive answer to HER2 detection challenges [1]. This scoring system has been the foundation for HER2 status determination for over two decades, guiding treatment decisions in breast cancer management. According to the 2018 ASCO/CAP guidelines, breast tumors are classified as HER2-positive when IHC is categorized as 3+ or shows ISH amplification, while HER2-negative is designated as 0, 1+, or 2+ with a negative ISH result (Figure 1, upper panel). Traditionally, the clinicopathological characteristics and survival outcomes of breast cancer patients are assessed by considering HER2 expression alongside estrogen and progesterone receptor status. Among breast cancer subtypes, triple-negative breast cancer (TNBC) is associated with the poorest overall and disease-free survival rates [14]. However, recent advancements in our understanding of HER2 expression have challenged this binary classification system. Recent studies have revealed substantial variation in HER2 expression within HER2-negative cases. This has led to the emergence of the “HER2-low” category (IHC1+ or 2+/ISH-negative), derived from previously classified HER2-negative tumors (Figure 1, lower panel) [3,8].

While both HER2-low and HER2-0 tumors are categorized as HER2-negative under traditional diagnostic criteria, accumulating evidence indicates meaningful biological and clinical differences between these subgroups. HER2-low tumors exhibit low but measurable HER2 protein expression without gene amplification, whereas HER2-0 tumors show no detectable HER2 expression. Additionally, HER2-low tumors are frequently hormone receptor-positive (HR+), although they can also occur within the TNBC subtype. In contrast, HER2-0 tumors are more frequently associated with TNBC [15].

As our understanding of HER2-low breast cancer continues to evolve, it is crucial to refine diagnostic methods and explore novel treatment approaches tailored to this newly recognized subset of patients.

## 3. Biological Characteristics of HER2-Low Breast Cancer

### 3.1. The Association of HER2-Low Breast Cancer with Hormone Receptors

HER2-low breast cancer is more frequently observed in patients with HR-positive breast cancer than in those with TNBC [3,14,16]. A complex relationship between HER2-low status and hormone receptor expression has been suggested. A meta-analysis found that HER2-low breast cancers show improved survival outcomes compared to HER2-0, with nuanced differences based on hormone receptor status, highlighting the complex interplay between HER2 and hormone receptor expression in prognosis and treatment response [17]. HER2-low tumors showed distinct patterns across hormone receptor subgroups. While HER2-low HR-positive cancers showed associations with younger age, higher HLA/pAKT, smaller tumors, and higher c-kit, HER2-low TNBC demonstrated the opposite patterns and was additionally linked to absent necrosis, higher pN stage, and lower CK14 [7]. These findings, reinforce that the features of HER2-low disease are primarily shaped by HR status.

In ER-positive HER2-negative breast cancer, further investigation is needed to understand the prognostic impact of HER2-low expression. Prior research indicates that the immune response—a significant prognostic factor in breast cancer—does not differ between the ER-positive and the ER-negative cohorts when comparing survival outcomes of HER2-low and HER2-positive patients [4]. Because ER status exerts a stronger prognostic influence than HER2, and because ER-low cancers demonstrate poorer outcomes than HER2-0 tumors, the correlation between HER2-low expression and ER levels may confound prognostic analyses; ER-low cases may disproportionately drive poorer survival signals attributed to HER2-low status [7,18]. Large pooled analyses further indicate that HER2-low status is associated with increased resistance to neoadjuvant chemotherapy in HR-positive breast cancer [19]. Consistently, ER-positive, HER2-low tumors show reduced chemotherapy sensitivity compared with ER-positive, HER2-negative disease [20]. Among ER-positive, HER2-low patients with residual disease after neoadjuvant treatment, factors such as high proliferation, low ER expression, and advanced stage both before and after therapy contribute to poorer prognosis [21]. These features provide valuable guidance for long-term therapeutic planning in this subgroup [21].

### 3.2. Molecular, Genomic, and Immunologic Features Across Subgroups

Integrative genomic and transcriptomic studies highlight substantial heterogeneity within HER2-low breast cancer, showing that HER2 IHC 1+ and 2+ tumors differ markedly depending on hormone-receptor (HR) status. For example, an integrative multi-omic analysis reported that IHC 1+ tumors displayed higher tumor mutational burden than IHC 2+ tumors, whereas equivocal IHC 2+ cases exhibited the greatest transcriptomic diversity [22]. A large cohort analysis (13,613 samples) further demonstrated that within HER2-low disease, HR-negative tumors harbored significantly more TP53 mutations, higher PD-L1 expression, and increased PIK3CA alterations compared with HR-positive HER2-low tumors [23]. Similarly, sequencing studies have shown that HR-positive HER2-low cancers are enriched for DNA damage–repair gene mutations, whereas HR-negative HER2-low cancers display prominent PI3K pathway alterations, which may underlie differential therapeutic responses and prognostic patterns [24]. Comparative genomic profiling consistently confirms that HR-positive HER2-low tumors resemble HR-positive HER2-0 tumors, while HR-negative HER2-low tumors cluster closely with HR-negative HER2-0 tumors, indicating that HER2-low is not a single biological entity but instead strongly shaped by HR status [25,26]. Notably, HER2-low TNBC exhibits distinct molecular and immune features compared with HER2-zero TNBC, including reduced immune activation, altered interferon signaling, and enrichment of epigenetic programs linked to immune evasion [27]. These findings support an intrinsically more immune-evasive phenotype in HER2-low TNBC and underscore the need for further clinical investigation.

## 4. Challenges in Establishing the Prognostic Value of HER2-Low

A growing body of evidence highlights substantial heterogeneity in the prognostic significance of HER2-low breast cancer, and a closer examination of individual studies helps explain why findings differ. In the Breast Invasive Carcinoma dataset (TCGA, PanCancer Atlas), analysis using cBioPortal suggests a potential, but non-significant trend toward longer survival in patients with ERBB2 alterations, consistent with prior studies on HER2-targeted therapies. Additionally, in the TCGA-BRCA and METABRIC cohorts, ERBB2 copy number variation (CNV) status was identified as an independent prognostic factor for relapse-free survival (RFS), with non-neutral CNV associated with improved outcomes [28].

Several investigations have reported that HER2-low tumors demonstrate more favorable long-term outcomes compared with HER2-zero disease. These studies observed reduced rates of pathological complete response (pCR) and recurrence score (RS), but with improved disease-free survival (DFS) and overall survival (OS) compared to HER2-0 cases [4,13,29,30,31,32]. Biological features may contribute to these observations: HER2-low tumors were shown to exhibit a lower immune response than HER2-zero tumors [4], and HER2 expression can evolve between primary tumors and residual disease (RD) after neoadjuvant therapy, reflecting underlying differences in tumor biology [13]. Additionally, HER2-low status has been associated with better breast cancer–specific survival in early-stage TNBC [29] and more favorable clinicopathological characteristics overall [30,31,32].

Conversely, a number of other large-scale and population-based studies found no significant differences in pCR, DFS, distant DFS, or OS between HER2-low and HER2-zero groups [3,16,33,34,35]. For example, analyses of national cancer registries [16] and meta-analyses in metastatic HR-positive disease treated with endocrine therapy plus cyclin-dependent kinases (CDKs) 4/6 inhibitors [35] suggest that treatment context and hormonal sensitivity may overshadow the prognostic contribution of HER2-low expression. Differences in cohort composition, molecular subtype distribution, treatment regimens, and analytical endpoints likely contribute to the conflicting outcomes reported across the literature.

Importantly, emerging data show that demographic and physiological factors modify the prognostic impact of HER2-low status. Racial differences, including disparities in HER2 expression distribution, treatment patterns, and comorbidities, were shown to influence outcomes in HER2-low disease [36]. Menopausal status was also found to significantly affect prognosis, particularly within the TNBC population, where postmenopausal HER2-low patients demonstrated distinct outcomes [32]. Age appears to be another important modifier: although HER2-low status alone did not serve as an independent prognostic marker in the overall cohort or within HR-positive and HR-negative subgroups, the combination of HER2-low status and younger age was associated with prognostic stratification specifically in TNBC [37]. Some recent reports further highlight that survival outcomes can be comparable between HER2-low and HER2-zero in certain early-stage cohorts, emphasizing that HER2-low biology does not confer a uniform survival pattern across all settings [38]. Conversely, HER2-zero disease has been associated with higher pCR rates than HER2-low disease in early-stage breast cancer receiving neoadjuvant therapy, suggesting potential differences in chemosensitivity that may contribute to divergent short-term and long-term outcomes [39].

Taken together, the variability in findings across studies, driven by differences in racial composition, age distribution, menopausal status, tumor subtype, immune microenvironment, and treatment exposure, helps explain the inconsistent prognostic conclusions reported for HER2-low breast cancer. These nuances highlight the need for deeper stratification in future studies to clarify the clinical implications of this emerging biological category.

## 5. Therapeutic Implications and Treatment Strategies for HER2-Low Disease

HER2 presents diverse mechanistic opportunities for drug development [11], including: (a) agents designed to block signaling activities, (b) agents delivering cytotoxic effectors to neoplastic cells, and (c) agents targeting the immune microenvironment [11]. Next-generation ADCs and technology-enabled precision oncology have revolutionized this therapeutic landscape, with emerging strategies such as trophoblast cell surface antigen 2 (TROP2)-targeted approaches demonstrating promising activity in preclinical studies [40].

In HER2-positive breast cancer cases, anti-HER2 therapies have improved outcomes. However, these treatments were traditionally unsuitable for the HER2-negative subtype, including those with low HER2 expression. Despite low HER2 expression in the HER2-low category, several targeted therapies have been proposed, such as ADCs, antibodies, HER2-derived vaccines, or small molecules targeting downstream signaling pathways (Figure 2). The landscape of HER2-low targeted therapies is rapidly evolving. A comprehensive review highlights the most promising approaches currently in clinical trials, including T-DXd, novel ADCs like SYD985 and RC48, bispecific antibodies such as zanidatamab, and combinations of HER2-targeted therapies with immunotherapy agents [13].

### Biologics Targeting HER2-Low Breast Cancers

Monoclonal antibodies represent another class of therapies against HER2-low breast cancer [11]. A notable example is MGAH22 (margetuximab), an Fc-engineered anti-HER2 monoclonal antibody designed to enhance immune activation and antibody-dependent cell-mediated cytotoxicity in HER2-positive breast cancer [44,45]. Bispecific antibodies (BsAbs) are another promising approach. These single protein molecules simultaneously identify two binding sites, facilitating the connection between immune cells and tumor cells to enhance anti-tumor responses. MCLA128 (zenocutuzumab) is an example of BsAb, specifically targeting both HER2 and HER3 receptors to directly inhibit tumor growth and, in combination with endocrine therapy, showed a disease control [46].

Third-generation antibody-drug conjugates (ADCs) have revolutionized HER2-low therapeutics, with trastuzumab deruxtecan’s January 2025 FDA approval for HER2-ultralow patients eliminating prior chemotherapy requirements. The global ADCs market, exceeding $12 billion, includes 17 approved conjugates and 2000+ in development, incorporating site-specific conjugation, novel cytotoxic payloads, and optimized drug-to-antibody ratios. Dual-payload conjugates and degrader-antibody conjugates advancing through clinical development promise enhanced efficacy through complementary mechanisms. Combination strategies with programmed cell death protein 1 (PD-1)/programmed death ligand 1 (PD-L1) inhibitors demonstrate synergistic anti-tumor effects, with trastuzumab deruxtecan plus pembrolizumab combinations showing promising early results [42,47].

Cancer vaccines represent another avenue for stimulating or amplifying anti-tumor immune responses in HER2-low breast cancer [48,49,50]. Recent advancements in vaccine design, such as the development of multi-epitope vaccines targeting both HER2 and other tumor-associated antigens, show promise for enhancing the efficacy of this approach in HER2-low breast cancer [51]. Combining HER2-targeted therapies with immune checkpoint inhibitors has also been shown to be promising [50]. Monoclonal antibodies targeting immune checkpoint receptors such as cytotoxic T lymphocyte-associated antigen (CTLA-4), PD-1, and PD-L have emerged as promising clinical targets [52,53].

It is important to note that conventional chemotherapy remains a viable option for HER2-low breast cancer, particularly as the first-line treatment in cases of aggressive, invasive disease with low ER expression [11]. While early studies did not show the benefits of adjuvant trastuzumab in HER2-low patients, numerous HER2-targeted therapies have demonstrated efficacy in treating HER2-low breast cancer. Appendix A provides an overview of some promising drugs in this space.

## 6. Challenges and Controversies Surrounding the Detection of HER2-Low Category

Accurately defining and monitoring HER2-low breast cancer has become a pivotal challenge in precision oncology. Although ADCs have created new therapeutic opportunities for patients with low levels of HER2 expression, the scientific community continues to debate how best to identify this subgroup. Biological complexity, technical variability, and evolving clinical criteria all contribute to uncertainty. The following sections outline key challenges and controversies from the inherent limits of tissue sampling and detection technologies to intratumoral heterogeneity and dynamic changes in HER2 status that collectively hinder reliable classification and optimal treatment planning.

### 6.1. Limitations of Tissue Biopsy

The limitations of current biopsy techniques, particularly core needle biopsy (CNB) and surgical excision specimens, in accurately determining HER2 status in HER2-low populations highlight the need for more advanced diagnostic approaches. [54]. Therefore, the diagnostic utility of CNB was constrained when it came to determining the HER2 status in breast cancer, particularly within the HER2-low population [55]. This limitation highlights the importance of exploring alternative diagnostic methods or refining the existing ones to improve the accuracy of HER2 status assessment, particularly in cases of HER2-low breast cancer. Further research and advancements in this area are essential to better serve patients and ensure they receive the most appropriate and effective treatment options based on their HER2 status. Emerging liquid biopsy techniques, such as circulating tumor DNA (ctDNA) analysis and extracellular vesicle (EVs) profiling, are showing potential in providing a more comprehensive and dynamic assessment of HER2 status, thereby aiming to overcome some key limitations of tissue biopsies [56].

### 6.2. Challenges in the Methods of HER2 Detection

The detection of HER2 expression in breast cancer, particularly for HER2-low tumors, presents several challenges [57,58]. IHC and ISH are the primary methods used, but both have significant limitations [59]. In this regard, IHC/ISH and quantitative reverse transcriptase-polymerase chain reaction (qRT-PCR) showed low agreement in estimating HER2 expression in HER2-negative tumors, suggesting that neither method is optimal for quantifying HER2-low expression [60,61].

Moreover, HER2 status often differs between CNB and surgical resection, with many HER2-0 cases reclassified as HER2-low in resections [62]. Interobserver variability is a significant issue in HER2 detection. Studies have shown low agreement among pathologists when grading HER2 IHC at low levels [60,63,64,65]. The ability of pathologists to achieve acceptable diagnostic accuracy in identifying patients with HER2-low breast cancer could be enhanced by short-term training [66]. Some studies have suggested complementary methods to improve HER2-low categorization. For instance, digital image analysis has been proposed as a reliable supplementary tool to enhance the standardization and quantification of HER2 IHC assessment, particularly in ambiguous cases with scores of 0 to 1+ [67]. The integration of AI represents the most transformative diagnostic breakthrough for HER2-low breast cancer in 2024–2025 [68,69,70,71]. In parallel, computational pathology techniques and digital imaging may also help to distinguish HER2-low from HER2-0 tumors more reliably [72]. Recent advancements in magnetic resonance imaging (MRI)-based radiomics and deep learning have demonstrated significant potential in the noninvasive differentiation of HER2 status in breast cancer [73,74,75,76,77,78,79]. However, the limited accessibility of these advanced technologies across many institutions highlights the need for practical and widely implementable methods.

While a definitive gold standard for HER2-low classification remains elusive, particularly for cases rated IHC 1+, the development of novel techniques is imperative. These advancements are crucial to enhance the accuracy of HER2-low detection, especially given the inadequacies of IHC/ISH in the context of novel antibody-drug conjugate treatments. Future research should focus on integrating multiple biomarkers and advanced technologies to create a more comprehensive and reliable classification system for HER2-low breast cancers.

### 6.3. Intra-Tumoral Heterogeneity of HER2 Expression

HER2 expression exhibits heterogeneity both within a single tumor and across metastatic sites, manifesting in various patterns such as clustered, mosaic, or scattered distributions. Recent studies have highlighted the importance of understanding this heterogeneity in HER2 expression, as it may impact treatment response and disease progression [80,81]. Intra-tumoral heterogeneity of HER2 is characterized by the presence of distinct subpopulations of tumor cells exhibiting differences within a primary tumor or between a primary tumor and its metastases, and is prognostically valuable [82]. This heterogeneity has been observed in up to 40% of breast cancer cases, with a notably higher prevalence in HER2-equivocal cases, while being rare in cases with a HER2 score of 3+ [83]. A study suggested that intra-tumoral heterogeneity and preanalytical factors can lead to variability in identifying HER2-low status between specimen types [84]. Recent studies using single-cell sequencing technologies have further elucidated the extent of this heterogeneity, revealing complex patterns of HER2 expression even within HER2-low tumors [85].

The ASCO/CAP addressed this issue in their 2009 and 2013 guidelines, defining HER2 heterogeneity as the presence of HER2 gene amplification in 5% to 50% of total tumor cells [86]. The 2018 guideline further acknowledged unusual patterns of HER2 expression, such as intense and complete staining in fewer than 10% of tumor cells, recommending retesting for these heterogeneous cancers upon recurrence and/or metastases due to the potential for changing HER2 status over time [1,87].

### 6.4. Conversion of HER2 Status

The dynamic nature of HER2 status in breast cancer has been well-documented, with numerous studies reporting conversions during treatment [88,89]. HER2 status has been shown to change during breast cancer progression in a significant proportion of cases, primarily shifting between HER2-0 and HER2-low categories [13,90,91]. A meta-analysis found that up to 30% of HER2-low tumors may change their HER2 status during disease progression, highlighting the need for repeated assessments [92].

Interestingly, HER2-positive breast cancer patients who experienced HER2 loss after neoadjuvant therapy did not transition to the HER2-low trait upon reaching RD [13,93]. These findings have several implications. The sensitivity of HER2-low expression to change following neoadjuvant therapy underscores the common occurrence of discrepancies in HR and/or HER2 status from primary tumor to RD [13]. Notably, over one-third of individuals initially characterized as HER2-0 exhibited HER2-low expression following neoadjuvant therapy, suggesting potential new therapeutic approaches for patients previously ineligible based on their primary tumor phenotype [13]. The transition from HER2-0 to HER2-low was more frequently observed in HR+ cancers compared to HR-negative tumors [91].

These findings highlight the importance of considering HER2 status as a continuum rather than a binary classification, especially in the context of neoadjuvant therapy and metastatic disease. Future research should focus on developing more sensitive and dynamic methods for HER2 detection and exploring the clinical implications of these status fluctuations in personalized breast cancer management.

## 7. Liquid Biopsy in the Clinical Management of HER2 Breast Cancer

Liquid biopsy analyzes circulating biomarkers, including circulating tumor cells (CTCs), ctDNA, and EVs, offering minimally invasive, real-time alternatives to tissue biopsy for HER2-low detection. This approach captures tumor heterogeneity comprehensively and enables dynamic monitoring of HER2 status changes, facilitating personalized treatment decisions with faster turnaround times than conventional tissue sampling (Figure 3) [94,95,96,97,98].

### 7.1. Circulating Tumor Cells or DNA

The diagnostic potential of ctDNA and CTCs in the management of HER2-positive breast cancer patients has been discussed previously [99,100,101]. The development of highly sensitive digital PCR techniques has enabled the detection of HER2 amplifications in plasma samples with greater accuracy [102,103].

D’Amico established a standardized pipeline for collecting and analyzing HER2-low CTCs, enhancing insights into their biological properties and predictive significance in breast cancer [104]. HER2 status discordance between the primary tumor and CTCs in approximately thirty percent of patients has been reported [105,106,107]. Notably, 32.1% of patients with histologically HER2-negative breast cancer had HER2-positive CTCs [105]. Although more studies are required, these findings emphasize the dynamic nature of HER2 expression and highlight the potential role of CTC analysis in refining treatment strategies for HER2-low breast cancer.

ctDNA analysis methods have demonstrated improved sensitivity in detecting HER2 amplifications, even in cases of HER2-low expression. The genomic landscape and prognostic significance of HER2-low using ctDNA in patients with metastatic breast cancer revealed that HER2-low did not represent a unique biologic subtype [8,108]. A retrospective cohort study using the commercially available Guardant360^®^ assay for ctDNA analysis showed that among patients with ER+ metastatic breast cancer (MBC), HER2-low tumors had a higher incidence of phosphatidylinositol-4,5-bisphosphate 3-kinase catalytic subunit alpha (*PIK3CA*) mutations and *MYC* amplification compared to HER2-0 MBC [109].

### 7.2. Circulating HER2 Extracellular Domain

HER2 is a 185 kDa transmembrane glycoprotein encoded by the HER2/neu gene, consisting of an extracellular domain (ECD), a lipophilic transmembrane domain, and an intracellular tyrosine kinase domain. However, unlike other members of the ERBB family, HER2 lacks a known ligand, making its activation mechanisms particularly intriguing. ECD adopts a constitutively open conformation, predisposing it to dimerization [110,111]. Activation occurs through homodimerization, heterodimerization with other ERBB receptors, and ECD shedding [112]. Notably, deletion of only 16 amino acids within the ECD completely abrogates HER2′s oncogenic potential, indicating the presence of a critical activating sequence that drives HER2-mediated oncogenic signaling (Figure 4) [113].

ECD shedding refers to the process in which the HER2-ECD is cleaved and released into the circulation [114]. The presence of elevated HER2 ECD levels in the serum of breast cancer patients has been extensively studied and proposed as a potential biomarker for identifying patients who might benefit from HER2-targeted therapies, monitoring treatment response, and early detection of disease recurrence or progression [112,115].

In 2003, the FDA approved the first enzyme-linked immunosorbent assay (ELISA)-based test for quantifying serum HER2 ECD levels, offering advantages over traditional tissue-based methods, such as non-invasiveness, real-time monitoring capabilities, quantitative results, and potential for detection of HER2 status changes [116]. Compared to IHC and ISH, measuring serum ECD by ELISA is a non-invasive method that provides real-time follow-up of patients. Besides, this method is quantitative and therefore can reduce intra- and interobserver variability for HER2 scoring.

Recent studies have further explored the utility of serum HER2 ECD as a biomarker in HER2-low breast cancer, with promising findings suggesting its potential as a complementary tool for identifying HER2-low tumors and predicting response to HER2-targeted therapies [117]. Given the potential clinical implications, a uniform HER2 cutoff for both genders may lead to misclassification and affect treatment decisions, highlighting the importance of gender-specific thresholds in HER2 assessment.

Serum HER2-ECD correlation with tissue expression varies significantly by disease stage and molecular subtype. While some studies have reported a correlation between tissue HER2 expression and serum HER2 ECD levels, many others have indicated no significant correlation, especially in early-stage primary breast cancer (Appendix A). Gender-specific thresholds may be necessary given higher circulating HER2-ECD levels in males versus females [118]. Despite these advancements, challenges remain in standardizing cut-off values and integrating this biomarker into clinical practice, necessitating further large-scale, prospective studies to validate its clinical utility in HER2-low breast cancer management.

## 8. The Next Wave of Liquid Biopsy Innovations

Although liquid biopsy provides a powerful, minimally invasive window into tumor biology, its full clinical impact, particularly in HER2-low disease, where biomarker signals are subtle—relies on integration with next-generation technologies. Advances such as AI, so-phisticated biosensors, multi-omic platforms, and emerging cellular messengers like extracellular vesicles can enhance sensitivity, enable real-time decentralized monitoring, and translate complex molecular data into actionable insights. These innovations (Figure 5), discussed in the following sections, are essential for transforming liquid biopsy from a promising tool into a routine component of precision breast cancer care.

### 8.1. AI–Enhanced Liquid Biopsy

AI is reshaping HER2-low breast cancer care by enhancing both tissue-based and liquid biopsy approaches. In pathology, AI improves the accuracy and consistency of IHC scoring, refines classification across the HER2 spectrum, and strengthens prediction of therapeutic response [71,119,120,121]. AI-based analysis can also identify more HER2-low and HER2-ultralow cases than conventional methods, revealing subtle expression changes across disease progression and increasing detection sensitivity [121]. A multi-laboratory investigation further demonstrated that AI-assisted HER2 interpretation provides higher inter-laboratory concordance and minimizes the subjective variability inherent in manual scoring, supporting its use in large-scale and multicenter studies [122]. Beyond tissue assessment, AI is driving major advances in liquid biopsy by integrating complex multi-omic data from circulating tumor DNA, extracellular vesicles, and circulating tumor cells, enabling more precise diagnosis, risk stratification, treatment selection, and monitoring of HER2-targeted therapeutic response. Additionally, AI can merge liquid-biopsy outputs with imaging, genomic, and clinical datasets to guide therapy choices, predict resistance, and support real-time adaptation of treatment plans as tumors evolve [123]. Despite this transformative potential, significant challenges remain—including issues of model generalizability, ethical and equity considerations, regulatory alignment, and the need for robust evidence and standardization—highlighting the importance of user trust, legal safeguards, and societal benefit [120,124,125]. Ultimately, more patient-tailored algorithms are required before AI can be fully integrated into routine precision care for HER2-low breast cancer [123].

### 8.2. Circulating EVs

EVs are heterogeneous, lipid bilayer membrane-delimited particles, released by all cell types, including tumor cells, and have introduced a new paradigm in understanding cellular communication [126,127]. Although tumor-derived EVs are present in low numbers in the bloodstream, they serve as messengers of tumor cells [128]. Developing diagnostic assays that target these vesicles could markedly enhance the accuracy of cancer detection. The greater abundance and stability of EVs compared to CTCs and ctDNA make them promising biomarkers for liquid biopsy [129,130]. The ability of EVs to provide real-time, comprehensive molecular profiles of tumors makes them particularly promising for personalized medicine approaches in breast cancer, potentially guiding treatment decisions and monitoring for the emergence of resistance mechanisms [131,132,133].

### 8.3. Biosensors and Point-of-Care (POC) Devices for HER2

Liquid biopsy has improved diagnostic accuracy by assessing ctDNA, CTCs, and EVs. However, these approaches still require complicated infrastructure, may not always provide quantitative data, and might lack the sensitivity needed for detecting HER2-low tumors. AI-integrated biosensor platforms represent revolutionary liquid biopsy technologies enabling real-time biomarker monitoring, blending biomedical engineering and oncology. Biosensors and POC devices are rapidly emerging as transformative tools in this context. By enabling fast, portable, and minimally invasive detection of HER2 from blood, serum, or saliva, these technologies could fundamentally shift HER2 testing from centralized laboratories to decentralized, patient-friendly settings, and ultimately toward POC monitoring.

Revolutionary biosensor platforms achieve clinical-grade sensitivity for detecting HER2 at very low levels: optical fiber-based sensors (151.5 attograms/mL in buffer, 3.7 picograms/mL in serum), gold electrode immunosensors (1 ng/mL detection), and liquid crystal biosensors (ultra-low 1 fg/mL detection limit). Multiplex electrochemical platforms simultaneously detect HER2 with other breast cancer biomarkers (0.5 ng/mL limit), enabling comprehensive molecular profiling from single blood drops [134,135,136,137,138].

Next-generation wearable biosensors represent the ultimate convergence of liquid biopsy and real-time monitoring. Revolutionary platforms integrate multiplex detection of HER2, ER, progesterone receptor, Ki-67, and ctDNA mutations from interstitial fluid or sweat, generating comprehensive molecular signatures from continuous sampling. The EpiView-D4 multimodal mobile pathology platform represents a paradigm shift toward accessible breast cancer diagnostics, combining brightfield cytology imaging with quantitative HER2 biomarker assessment on a smartphone-based system. This dual-function platform achieves a 77 pM detection limit for HER2 protein in cellular lysates, with only 3.6-fold lower sensitivity compared to laboratory-grade fluorescence scanners [134,135,138].

While challenges remain, including regulatory approval, large-scale validation, reproducibility across populations, affordability, and integration with electronic health record systems, the current trajectory suggests that near-to-patient multiplex HER2 biosensors may soon transition from concept to reality. Such innovations could significantly enhance early detection, monitoring, and personalized management of breast cancer, with particular relevance for HER2-low disease, where subtle biomarker changes can influence treatment choice and long-term outcomes. Figure 6 illustrates the continuum of breast cancer diagnostics, from liquid biopsy and lab-based testing to lab-on-a-chip and POC devices. These emerging platforms highlight the shift toward POC testing for early diagnosis and prognosis.

## 9. Critical Clinical Questions and Technology-Driven Solutions for HER2-Low Breast Cancer

The recognition of HER2-low breast cancer has not only transformed a biological concept into a clinically meaningful category but has also opened the door to a new era of therapeutic possibilities. These questions define the next decade of HER2-low breast cancer management, requiring interdisciplinary collaboration between oncologists, pathologists, bioengineers, and data scientists to deliver truly personalized cancer care:**Precision medicine integration:** How can integrated AI-assisted pathology, liquid biopsy, and genomic profiling platforms create comprehensive HER2 status assessments enabling real-time treatment decisions?**Therapeutic resistance mechanisms:** What novel ADC combination strategies overcome resistance in HER2-low breast cancer, and how can liquid biopsy monitor resistance emergence for rapid therapeutic pivoting?**Technology democratization:** How can POC biosensors and AI-assisted diagnostics eliminate disparities in HER2-low detection and treatment access, particularly in resource-limited settings?**Biomarker integration:** What molecular signatures beyond HER2 expression (tumor-infiltrating lymphocytes, homologous recombination deficiency, immune profiles) can further stratify HER2-low patients for optimal therapeutic selection?**Wearable monitoring revolution:** How can healthcare systems integrate continuous liquid biopsy monitoring with wearable biosensor technologies, enabling early resistance detection and dynamic treatment optimization?

## 10. Conclusions

This review highlights the emerging significance of HER2-low breast cancer as a clinically relevant entity, despite challenges in its accurate classification and assessment. The current gold standard for evaluating HER2 status, involving ISH tests in conjunction with IHC assays, faces limitations in distinguishing between low HER2 levels and the absence of HER2 expression. Challenges persist due to the semi-quantitative nature of IHC, its limited sensitivity in detecting very low amounts of HER2 protein, and various factors affecting analysis outcomes. The complex relationship between HER2 and hormone receptor status, intra-tumoral heterogeneity of HER2 expression, and conversion of HER2 status further complicates the landscape. In our recent publication, we presented a Comprehensive Oncological Biomarker Framework that integrates genetic and molecular testing, imaging, histopathology, multi-omics, and liquid biopsy to generate a unique molecular fingerprint for each patient [139]. AI-driven precision diagnostics and digital pathology transformation represent revolutionary liquid biopsy technologies for improving the accuracy of HER2-low breast cancer diagnosis and monitoring. Future research directions should focus on developing more reliable approaches to assess low HER2 expression, conducting further clinical trials stratified for HER2-low breast cancer, and refining our understanding of the interplay between HER2-low status and hormone receptor expression.

Despite these challenges, recent clinical trials have demonstrated improved treatment outcomes by targeting HER2-low expression, particularly with ADCs showing promising efficacy in advanced disease. Emerging techniques offer new hope in addressing these challenges and improving HER2-low breast cancer management. The ability of liquid biopsy to provide real-time, comprehensive molecular profiles of tumors makes it particularly promising for personalized medicine approaches in breast cancer, potentially guiding treatment decisions and monitoring for the emergence of resistance mechanisms. While significant progress has been made, further research is crucial to fully realize the diagnostic and therapeutic potential of HER2-low breast cancer, advancing towards more precise and effective cancer management strategies in the era of personalized medicine.

## Figures and Tables

**Figure 1 biomedicines-14-00049-f001:**
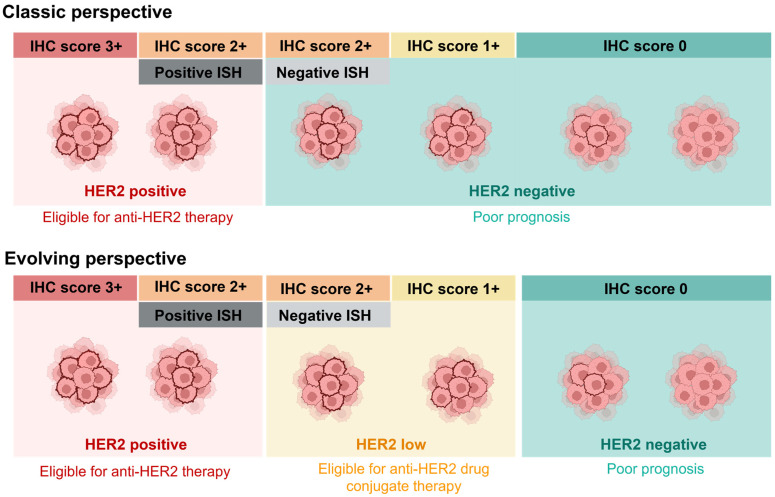
Classification of breast cancer based on human epidermal growth factor receptor 2 (HER2) expression. Upper panel: The classic perspective categorizes breast cancers as either HER2-negative (immunohistochemistry (IHC) 0, 1+, or 2+ with negative in situ hybridization (ISH)) or HER2-positive (IHC 3+ or ISH-positive), determining eligibility for anti-HER2 therapy. Lower panel: The evolving perspective introduces a new “HER2-low” category (IHC 1+ or 2+ with negative ISH), which is derived from tumors previously classified as HER2-negative. This new classification highlights the potential for targeted therapies in a subset of tumors previously considered ineligible for HER2-directed treatments. Created in BioRender. Salahandish, R. (2025) https://BioRender.com/w7dhlp4 (accessed on 19 November 2025).

**Figure 2 biomedicines-14-00049-f002:**
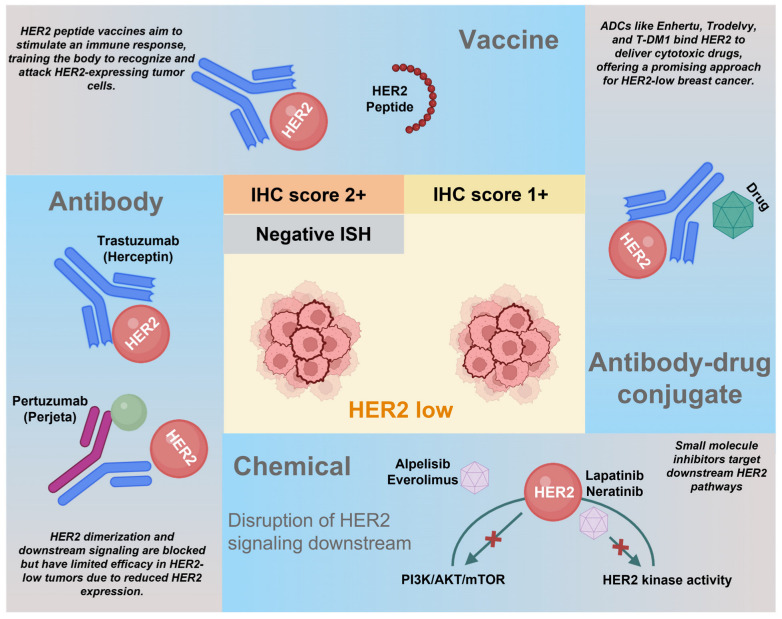
Treatment strategies for HER2-low breast cancer. Notwithstanding the low expression of HER2 within the HER2-low category, a multitude of HER2-targeted therapies have been proposed. These encompass a range of approaches, such as antibody-drug conjugates (ADCs), antibodies, HER2-derived vaccines, and chemicals targeting HER2 downstream signaling. AKT, Protein kinase B; HER2, Human epidermal growth factor receptor 2; IHC, Immunohistochemistry; ISH, In-situ hybridization; mTOR, Mammalian target of rapamycin; PI3K, Phosphoinositide 3-kinase. DESTINY-Breast04 demonstrated T-DXd’s transformative efficacy in HER2-low breast cancer, leading to FDA approval in August 2022. Ongoing ADC development includes 17 approved conjugates globally with 2000+ in clinical development, representing a $12+ billion market focused on site-specific conjugation and novel cytotoxic payloads [11,41,42] (summarized in Appendix A). A recent systematic review and meta-analysis confirm the efficacy of ADC in HER2-low advanced/metastatic breast cancer patients over treatment of the physician’s choice [43]. Created in BioRender. Salahandish, R. (2025) https://BioRender.com/w7dhlp4 (accessed on 19 November 2025).

**Figure 3 biomedicines-14-00049-f003:**
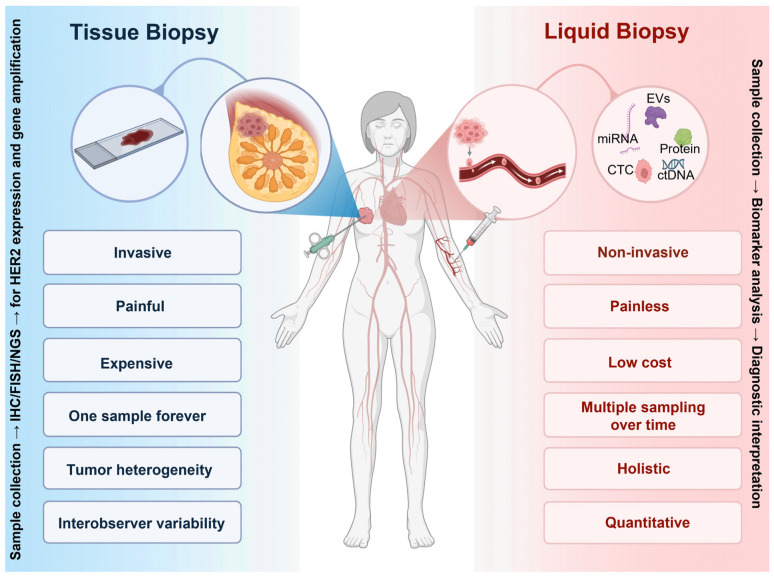
Tissue biopsy vs. liquid biopsy for HER2-low breast cancer diagnosis. Tissue biopsy (left panel) is the approved method of biomarker detection; however, it has some limitations and disadvantages. Liquid biopsy (right panel) and analyzing circulating biomarkers, including circulating tumor cells (CTCs); circulating tumor DNA (ctDNA), and extracellular vesicles (EVs), is not an approved method; however, it has some advantages that make it promising for future developments. Created in BioRender. Salahandish, R. (2025) https://BioRender.com/w7dhlp4 (accessed on 19 November 2025).

**Figure 4 biomedicines-14-00049-f004:**
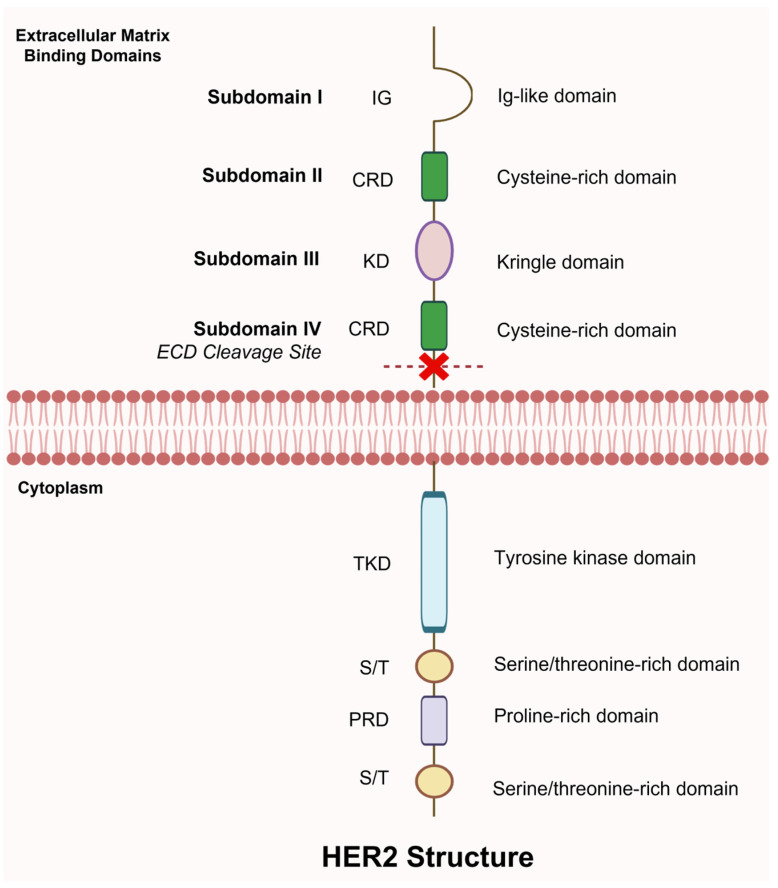
Schematic diagram representing the HER2 extracellular domain (ECD). The ECD is a 185 kDa fragment shed into circulation and consists of four subdomains(I–IV), each playing a role in receptor activation and dimerization. Subdomain I (Ig-like domain) participates in ligand binding, Subdomains II and III (cysteine-rich and Kringle domains) are crucial for heterodimerization, and Subdomain IV (cysteine-rich domain) is the site of proteolytic cleavage. Created in BioRender. Salahandish, R. (2025) https://BioRender.com/w7dhlp4 (accessed on 19 November 2025).

**Figure 5 biomedicines-14-00049-f005:**
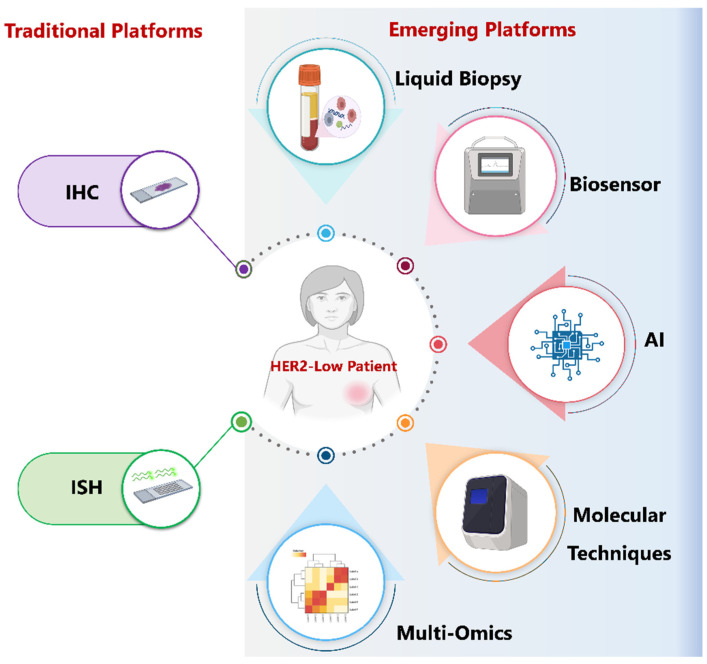
Traditional and emerging platforms for HER2-low breast cancer assessment. This schematic contrasts conventional tissue-based methods—immunohistochemistry (IHC) and in situ hybridization (ISH)—with emerging technologies that enhance the detection and characterization of HER2-low disease. Liquid biopsy, biosensors, molecular techniques, and multi-omics approaches offer more sensitive and comprehensive evaluation beyond tissue analysis. Artificial intelligence (AI) further supports these platforms by refining HER2 quantification, reducing observer variability, and integrating multi-modal data to guide personalized management of HER2-low breast cancer.

**Figure 6 biomedicines-14-00049-f006:**
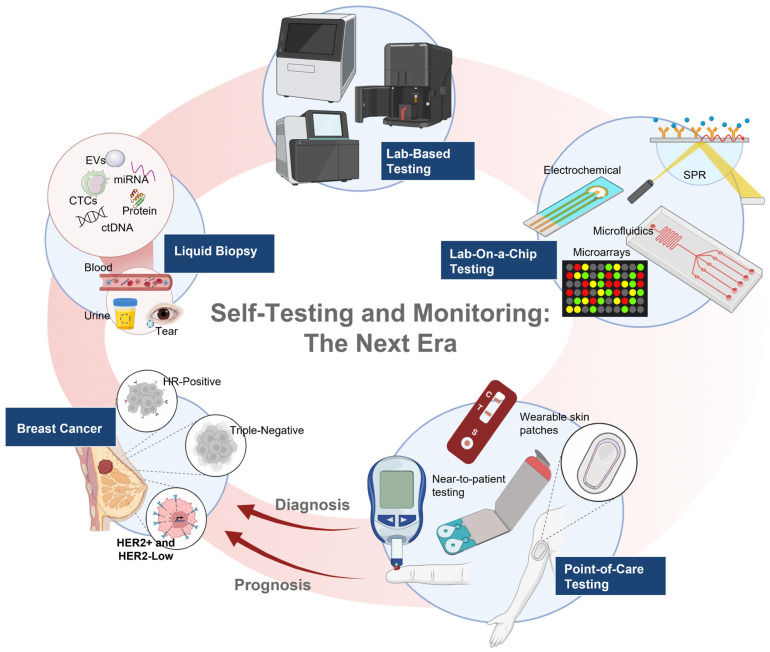
Future directions for breast cancer diagnostics and monitoring. The workflow illustrates how liquid biopsy approaches (detecting circulating tumor DNA (ctDNA), circulating tumor cells (CTCs), extracellular vesicles (EVs), proteins, and miRNAs from biofluids such as blood, urine, and tears) are integrated with conventional lab-based testing and advanced lab-on-a-chip platforms (microfluidics, electrochemical sensors, surface plasmon resonance (SPR), and microarrays). These advances are converging into point-of-care solutions, including at-home testing devices and wearable skin patches, with the potential to transform diagnosis, prognosis, and monitoring of breast cancer subtypes, particularly HER2-low disease, by enabling accessible, real-time assessment outside the laboratory. Created in BioRender. Salahandish, R. (2025) https://BioRender.com/w7dhlp4 (accessed on 19 November 2025).

## Data Availability

No new data were created or analyzed in this study. Data sharing does not apply to this article.

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
