# Peer review of "HER2-Low Breast Cancer at the Interface of Pathology and Technology: Toward Precision Management"

_biomedicines, 2025, doi:10.3390/biomedicines14010049_

Round 1
Reviewer 1 Report
Comments and Suggestions for Authors
This manuscript presents a comprehensive overview of the emerging and critically important subject of HER2-low breast cancer, covering pathology, clinical oncology, diagnostics, liquid biopsy advances, artificial intelligence, and therapeutic strategies, while providing valuable future directions. The topic is highly relevant given the rapidly evolving landscape of HER2-low classification and its implications for antibody–drug conjugates (ADCs). The review is supported by a wide range of literature. However, the manuscript requires significant structural and organizational revisions to improve clarity and coherence, including:
- The manuscript jumps between topics (e.g., from therapeutics directly to detection challenges, then to liquid biopsy). A more logical flow is needed. A suggested structure could be: Introduction → Definition/Classification → Clinical/Biological Characteristics → Current Detection Challenges → Emerging Technological Solutions (Liquid Biopsy, AI, Biosensors) → Therapeutics → Future Directions/Conclusions.
- Limitations of immunohistochemistry (IHC) and in-situ hybridization (ISH), together with the definition of HER2-low, are repeated in multiple sections (e.g., Introduction, Section 2, Section 5). This should be consolidated.
- Section 3 (Association with Hormone Receptors) is a crucial topic; yet, it is brief and requires further elaboration.
- The authors should offer a hypothesis to explain the conflicting data on prognosis (Page 3).
- The recent FDA approval for HER2-ultralow requires clarification of how this differs from HER2-low.
Author Response
- Limitations of immunohistochemistry (IHC) and in-situ hybridization (ISH), together with the definition of HER2-low, are repeated in multiple sections (e.g., Introduction, Section 2, Section 5). This should be consolidated.
Response: Thank you for your valuable feedback. In response to this and the previous comment, we have revised the manuscript structure to ensure a more logical flow of topics and to eliminate redundancies.
- Section 3 (Association with Hormone Receptors) is a crucial topic; yet, it is brief and requires further elaboration.
Response: We appreciate your valuable comment and added more details with some new references to this part as below:
3.1. The Association of HER2-Low Breast Cancer with Hormone Receptors
HER2-low breast cancer is more frequently observed in patients with HR-positive breast cancer than in those with TNBC [3,15,17]. A complex relationship between HER2-low status and hormone receptor expression has been suggested. A meta-analysis found that HER2-low breast cancers show improved survival outcomes compared to HER2-0, with nuanced differences based on hormone receptor status, highlighting the complex interplay between HER2 and hormone receptor expression in prognosis and treatment response [18]. HER2-low tumors showed distinct patterns across hormone receptor subgroups. While HER2-low HR-positive cancers showed associations with younger age, higher HLA/pAKT, smaller tumors, and higher c-kit, HER2-low TNBC demonstrated the opposite patterns and was addi-tionally linked to absent necrosis, higher pN stage, and lower CK14 [19]. These findings, reinforce that the features of HER2-low disease are primarily shaped by HR status.
In ER-positive HER2-negative breast cancer, further investigation is needed to understand the prognostic impact of HER2-low expression. Prior research indicates that the immune response—a significant prognostic factor in breast cancer—does not differ between the ER-positive and the ER-negative cohorts when comparing survival outcomes of HER2-low and HER2-positive patients [4]. Because ER status exerts a stronger prognostic influence than HER2, and because ER-low cancers demonstrate poorer outcomes than HER2-0 tumors, the correlation between HER2-low expression and ER levels may confound prognostic analyses; ER-low cases may disproportionately drive poorer survival signals attributed to HER2-low status [7,20]. Large pooled analyses further indicate that HER2-low status is associated with increased re-sistance to neoadjuvant chemotherapy in HR-positive breast cancer [21]. Consistently, ER-positive, HER2-low tumors show reduced chemotherapy sensitivity compared with ER-positive, HER2-negative disease [22]. Among ER-positive, HER2-low patients with residual disease after neoadjuvant treatment, factors such as high proliferation, low ER expression, and advanced stage both before and after therapy contribute to poorer prognosis [23]. These features provide valuable guidance for long-term therapeutic planning in this subgroup.
- The authors should offer a hypothesis to explain the conflicting data on prognosis (Page 3).
Response: We appreciate the reviewer’s comment. We have now added a dedicated section addressing this point and explaining the conflicting prognostic data, as shown below.
- Challenges in Establishing the Prognostic Value of HER2-Low
A growing body of evidence highlights substantial heterogeneity in the prognostic significance of HER2-low breast cancer, and a closer examination of individual studies helps explain why findings differ. In the Breast Invasive Carcinoma dataset (TCGA, PanCancer Atlas), analysis using cBioPortal suggests a potential, but non-significant trend toward longer survival in patients with ERBB2 altera-tions, consistent with prior studies on HER2-targeted therapies. Additionally, in the TCGA-BRCA and METABRIC cohorts, ERBB2 copy number variation (CNV) status was identified as an independent prognostic factor for relapse-free survival (RFS), with non-neutral CNV associated with improved out-comes [30].
Several investigations have reported that HER2-low tumors demonstrate more favorable long-term outcomes compared with HER2-zero disease. These studies observed reduced rates of pathological complete response (pCR) and recurrence score (RS), but with improved disease-free survival (DFS) and overall survival (OS) compared to HER2-0 cases [4,14,31-34]. Biological features may contribute to these observations: HER2-low tumors were shown to exhibit a lower immune response than HER2-zero tumors [4] , and HER2 expression can evolve between primary tumors and residual disease (RD) after neoadjuvant therapy, reflecting underlying differences in tumor biology [14] . Additionally, HER2-low status has been associated with better breast cancer–specific survival in early-stage TNBC [31] and more favorable clinicopathological characteristics overall [32-34].
Conversely, a number of other large-scale and population-based studies found no significant differences in pCR, DFS, distant DFS, or OS between HER2-low and HER2-zero groups [3,17,35-37]. For example, analyses of national cancer registries [17] and meta-analyses in metastatic HR-positive disease treated with endocrine therapy plus cyclin-dependent kinases (CDKs) 4/6 inhibitors [37] suggest that treatment context and hormonal sensitivity may overshadow the prognostic contribution of HER2-low expression. Differences in cohort composition, molecular subtype distribution, treatment regimens, and analytical endpoints likely contribute to the conflicting outcomes reported across the literature.
Importantly, emerging data show that demographic and physiological factors modify the prognostic impact of HER2-low status. Racial differences, including disparities in HER2 expression distribution, treatment patterns, and comorbidities, were shown to influence outcomes in HER2-low disease [38] . Menopausal status was also found to significantly affect prognosis, particularly within the TNBC population, where postmenopausal HER2-low patients demonstrated distinct outcomes [34]. Age ap-pears to be another important modifier: although HER2-low status alone did not serve as an independent prognostic marker in the overall cohort or within HR-positive and HR-negative subgroups, the com-bination of HER2-low status and younger age was associated with prognostic stratification specifically in TNBC [39]. Some recent reports further highlight that survival outcomes can be comparable between HER2-low and HER2-zero in certain early-stage cohorts, emphasizing that HER2-low biology does not confer a uniform survival pattern across all settings [40]. Conversely, HER2-zero disease has been associated with higher pCR rates than HER2-low disease in early-stage breast cancer receiving neoad-juvant therapy, suggesting potential differences in chemosensitivity that may contribute to divergent short-term and long-term outcomes [41].
Taken together, the variability in findings across studies, driven by differences in racial composition, age distribution, menopausal status, tumor subtype, immune microenvironment, and treatment exposure, helps explain the inconsistent prognostic conclusions reported for HER2-low breast cancer. These nuances highlight the need for deeper stratification in future studies to clarify the clinical implications of this emerging biological category.
- The recent FDA approval for HER2-ultralow requires clarification of how this differs from HER2-low.
Response: Thanks for your comment. We re-wrote the sentence to be complete and clear, and added a reference as below:
Recently, the U.S. Food and Drug Administration (FDA) approved trastuzumab deruxtecan (T-DXd, DS-8201a) for patients with unresectable or metastatic breast cancer who have progressed on prior endocrine therapy and have hormone receptor–positive tu-mors classified as HER2-low (IHC 1+ or IHC 2+/ISH–) or HER2-ultralow (IHC 0 with detect-able membrane staining) [50].
Reviewer #2
The manuscript focus on an important topic: HER2-low breast cancer. However, I suggest to take into account several changes to be more depth, clarity, and specific to journal sstandars. Please consider the following suggestions:
- Abstract: It is not following the structured format. Please check the Biomedicines requirements and re-write according to this items.
Response: Thanks for bringing it to our attention. We updated the abstract based on journal requirements as follows:
Background: HER2-low breast cancer has emerged as a clinically meaningful category that challenges the historical HER2-positive versus HER2-negative classification. Although not defined as a distinct biological subtype, HER2-low tumors exhibit unique clinicopathological features and differential sensitivity to novel antibody–drug conjugates. Accurate identification remains difficult due to limitations in immunohistochemistry performance, inter-observer variability, intratumoral heterogeneity, and dynamic shifts in HER2 expression over time.
Objectives: This review synthesizes current evidence on the biological and clinical characteristics of HER2-low breast cancer and evaluates emerging diagnostic innovations, with emphasis on liquid biopsy approaches and evolving technologies that may enhance diagnostic accuracy and monitoring.
Methods: A narrative literature review was conducted, examining tissue-based HER2 testing, liquid biopsy modalities—including circulating tumor cells, circulating nucleic acids, extracellular vesicles, and soluble HER2 extracellular domains—and applications of artificial intelligence (AI) across histopathology and multimodal diagnostic systems.
Results: Liquid biopsy technologies offer minimally invasive, real-time assessment of HER2 dynamics and may overcome fundamental limitations of tissue-based assays. However, these platforms require rigorous analytical validation and face regulatory and standardization challenges before widespread clinical adoption. Concurrently, AI-enhanced histopathology and multimodal diagnostic systems improve reproducibility, refine HER2 classification, and enable more accurate prediction of treatment response. Emerging biosensor- and AI-enabled monitoring frameworks further support continuous disease evaluation.
Conclusions: HER2-low breast cancer sits at the intersection of evolving pathology and technological innovation. Integrating liquid biopsy platforms with AI-driven diagnostics has the potential to advance precision stratification and guide personalized therapeutic strategies for this expanding patient subgroup.
- Lines 118-129: I suggest to provide more detail from the studies you reffered it to explain why there are differences between studies and how the demographic and physiology is affecting.
Response: We have now added a dedicated section addressing this point and explaining the conflicting prognostic data, as shown below:
- Challenges in Establishing the Prognostic Value of HER2-Low
A growing body of evidence highlights substantial heterogeneity in the prognostic significance of HER2-low breast cancer, and a closer examination of individual studies helps explain why findings differ. In the Breast Invasive Carcinoma dataset (TCGA, PanCancer Atlas), analysis using cBioPortal suggests a potential, but non-significant trend toward longer survival in patients with ERBB2 altera-tions, consistent with prior studies on HER2-targeted therapies. Additionally, in the TCGA-BRCA and METABRIC cohorts, ERBB2 copy number variation (CNV) status was identified as an independent prognostic factor for relapse-free survival (RFS), with non-neutral CNV associated with improved out-comes [30].
Several investigations have reported that HER2-low tumors demonstrate more favorable long-term outcomes compared with HER2-zero disease. These studies observed reduced rates of pathological complete response (pCR) and recurrence score (RS), but with improved disease-free survival (DFS) and overall survival (OS) compared to HER2-0 cases [4,14,31-34]. Biological features may contribute to these observations: HER2-low tumors were shown to exhibit a lower immune response than HER2-zero tumors [4] , and HER2 expression can evolve between primary tumors and residual disease (RD) after neoadjuvant therapy, reflecting underlying differences in tumor biology [14] . Additionally, HER2-low status has been associated with better breast cancer–specific survival in early-stage TNBC [31] and more favorable clinicopathological characteristics overall [32-34].
Conversely, a number of other large-scale and population-based studies found no significant differences in pCR, DFS, distant DFS, or OS between HER2-low and HER2-zero groups [3,17,35-37]. For example, analyses of national cancer registries [17] and meta-analyses in metastatic HR-positive disease treated with endocrine therapy plus cyclin-dependent kinases (CDKs) 4/6 inhibitors [37] suggest that treatment context and hormonal sensitivity may overshadow the prognostic contribution of HER2-low expression. Differences in cohort composition, molecular subtype distribution, treatment regimens, and analytical endpoints likely contribute to the conflicting outcomes reported across the literature.
Importantly, emerging data show that demographic and physiological factors modify the prognostic impact of HER2-low status. Racial differences, including disparities in HER2 expression distribution, treatment patterns, and comorbidities, were shown to influence outcomes in HER2-low disease [38] . Menopausal status was also found to significantly affect prognosis, particularly within the TNBC population, where postmenopausal HER2-low patients demonstrated distinct outcomes [34]. Age ap-pears to be another important modifier: although HER2-low status alone did not serve as an independent prognostic marker in the overall cohort or within HR-positive and HR-negative subgroups, the com-bination of HER2-low status and younger age was associated with prognostic stratification specifically in TNBC [39]. Some recent reports further highlight that survival outcomes can be comparable between HER2-low and HER2-zero in certain early-stage cohorts, emphasizing that HER2-low biology does not confer a uniform survival pattern across all settings [40]. Conversely, HER2-zero disease has been associated with higher pCR rates than HER2-low disease in early-stage breast cancer receiving neoad-juvant therapy, suggesting potential differences in chemosensitivity that may contribute to divergent short-term and long-term outcomes [41].
Taken together, the variability in findings across studies, driven by differences in racial composition, age distribution, menopausal status, tumor subtype, immune microenvironment, and treatment exposure, helps explain the inconsistent prognostic conclusions reported for HER2-low breast cancer. These nuances highlight the need for deeper stratification in future studies to clarify the clinical implications of this emerging biological category.
- Lines 154-161: I would suggest examples from the cited studies (7, 28) and explain in more details how ER status is influencing in HER2-low breast cancer.
Response: We appreciate your valuable comment and added more details with some new references to this part as below:
3.1. The Association of HER2-Low Breast Cancer with Hormone Receptors
HER2-low breast cancer is more frequently observed in patients with HR-positive breast cancer than in those with TNBC [3,15,17]. A complex relationship between HER2-low status and hormone receptor expression has been suggested. A meta-analysis found that HER2-low breast cancers show improved survival outcomes compared to HER2-0, with nuanced differences based on hormone receptor status, highlighting the complex interplay between HER2 and hormone receptor expression in prognosis and treatment response [18]. HER2-low tumors showed distinct patterns across hormone receptor subgroups. While HER2-low HR-positive cancers showed associations with younger age, higher HLA/pAKT, smaller tumors, and higher c-kit, HER2-low TNBC demonstrated the opposite patterns and was addi-tionally linked to absent necrosis, higher pN stage, and lower CK14 [19]. These findings, reinforce that the features of HER2-low disease are primarily shaped by HR status.
In ER-positive HER2-negative breast cancer, further investigation is needed to understand the prognostic impact of HER2-low expression. Prior research indicates that the immune response—a significant prognostic factor in breast cancer—does not differ between the ER-positive and the ER-negative cohorts when comparing survival outcomes of HER2-low and HER2-positive patients [4]. Because ER status exerts a stronger prognostic influence than HER2, and because ER-low cancers demonstrate poorer outcomes than HER2-0 tumors, the correlation between HER2-low expression and ER levels may confound prognostic analyses; ER-low cases may disproportionately drive poorer survival signals attributed to HER2-low status [7,20]. Large pooled analyses further indicate that HER2-low status is associated with increased re-sistance to neoadjuvant chemotherapy in HR-positive breast cancer [21]. Consistently, ER-positive, HER2-low tumors show reduced chemotherapy sensitivity compared with ER-positive, HER2-negative disease [22]. Among ER-positive, HER2-low patients with residual disease after neoadjuvant treatment, factors such as high proliferation, low ER expression, and advanced stage both before and after therapy contribute to poorer prognosis [23]. These features provide valuable guidance for long-term therapeutic planning in this subgroup.
- Lines 266-267: Please provide details on how AI improves HER2 quantification.
Response: Thank you for this insightful comment. We have now expanded the manuscript to provide specific details on how AI improves HER2 quantification. The revised text as follows:
AI is reshaping HER2-low breast cancer care by enhancing both tissue-based and liquid biopsy approaches. In pathology, AI improves the accuracy and consistency of IHC scoring, refines classi-fication across the HER2 spectrum, and strengthens prediction of therapeutic response [118-121]. AI-based analysis can also identify more HER2-low and HER2-ultralow cases than conventional methods, revealing subtle expression changes across disease progression and increasing detection sensitivity [121]. A multi-laboratory investigation further demonstrated that AI-assisted HER2 in-terpretation provides higher inter-laboratory concordance and minimizes the subjective variability inherent in manual scoring, supporting its use in large-scale and multicenter studies [122]. Beyond tissue assessment, AI is driving major advances in liquid biopsy by integrating complex multi-omic data from circulating tumor DNA, extracellular vesicles, and circulating tumor cells, enabling more precise diagnosis, risk stratification, treatment selection, and monitoring of HER2-targeted therapeutic response. Additionally, AI can merge liquid-biopsy outputs with imaging, genomic, and clinical datasets to guide therapy choices, predict resistance, and support real-time adaptation of treatment plans as tumors evolve [123]. Despite this transformative potential, significant challenges remain—including issues of model generalizability, ethical and equity considerations, regulatory alignment, and the need for robust evidence and standardization—highlighting the importance of user trust, legal safeguards, and societal benefit [120,124,125]. Ultimately, more patient-tailored algorithms are required before AI can be fully integrated into routine precision care for HER2-low breast cancer [123].
- Lines 319-321: I suggest to give more data to understand why you confirm this
Response: Thanks for your valuable comment. We added new dedicated subsection with new details as bellow:
3.2. Molecular, Genomic, and Immunologic Features Across Subgroups
Integrative genomic and transcriptomic studies highlight substantial heterogeneity within HER2-low breast cancer, showing that HER2 IHC 1+ and 2+ tumors differ markedly depending on hor-mone-receptor (HR) status. For example, an integrative multi-omic analysis reported that IHC 1+ tumors displayed higher tumor mutational burden than IHC 2+ tumors, whereas equivocal IHC 2+ cases ex-hibited the greatest transcriptomic diversity [24]. A large cohort analysis (13,613 samples) further demonstrated that within HER2-low disease, HR-negative tumors harbored significantly more TP53 mutations, higher PD-L1 expression, and increased PIK3CA alterations compared with HR-positive HER2-low tumors [25]. Similarly, sequencing studies have shown that HR-positive HER2-low cancers are enriched for DNA damage–repair gene mutations, whereas HR-negative HER2-low cancers display prominent PI3K pathway alterations, which may underlie differential therapeutic responses and prog-nostic patterns [26]. Comparative genomic profiling consistently confirms that HR-positive HER2-low tumors resemble HR-positive HER2-0 tumors, while HR-negative HER2-low tumors cluster closely with HR-negative HER2-0 tumors, indicating that HER2-low is not a single biological entity but instead strongly shaped by HR status [27,28]. Notably, HER2-low TNBC exhibit distinct molecular and immune features compared with HER2-zero TNBC, including reduced immune activation, altered in-terferon signaling, and enrichment of epigenetic programs linked to immune evasion [29]. These findings support an intrinsically more immune-evasive phenotype in HER2-low TNBC and underscore the need for further clinical investigation.
Reviewer #3
The paper addresses the important clinical issue of Her 2 low breast cancer and outlines the significant clinical research with multiple antibody drug conjugates in trials as they outline and catalogue. The topic is highly clinically relevant as many clinicians struggle to determine her 2 low eligibility for patients - for clinicians the paper highlights the multiple available platforms in this area and explores evolving new technologies ranging from ctDNA to ev to digital wearable devices. The paper is comprehensive integrating clinical and lab based data and platforms in this field and summarises ongoing clinical trials.
- The paper appears comprehensive but as 50% of the references are not in the reference list it is hard to judge this completely. In supplement 1 one study is cited with not available beside the data - can the authors clarify ?
Response: Thank you for bringing this to our attention. We have carefully revised and updated the reference list in the revised version of our manuscript.
- The paper is well written and comprehensive - it would be helped by more figures to make it less text dense - i would suggest a figure outlining the number of platforms used for her 2 assessment at present and ones under development; and i feel a figure similar to what the authors outline they have done for reference 118 in the paper to pull the themes mentioned together.
Response: Thank you for your valuable comment. To address it, we have prepared the following figure in the revised version of our manuscript.
Please see the attachment.

Reviewer 2 Report
Comments and Suggestions for Authors
The manuscript focus on an important topic: HER2-low breast cancer. However, I suggest to take into account several changes to be more depth, clarity, and specific to journal sstandars. Please consider the following suggestions:
Abstract: It is not following the structured format. Please check the Biomedicines requirements and re-write according to this items.
Lines 118-129: I suggest to provide more detail from the studies you reffered it to explain why there are differences between studies and how the demographic and physiology is affecting.
Lines 154-161: I would suggest examples from the cited studies (7, 28) and explain in more details how ER status is influencing in HER2-low breast cancer.
Lines 266-267: Please provide details on how AI improves HER2 quantification.
Lines 319-321: I suggest to give more data to understand why you confirm this
Reviewer 3 Report
Comments and Suggestions for Authors
The paper addresses the important clinical issue of Her 2 low breast cancer and outlines the significant clinical research with multiple antibody drug conjugates in trials as they outline and catalogue. The topic is highly clinically relevant as many clinicians struggle to determine her 2 low eligibility for patients - for clinicians the paper highlights the multiple available platforms in this area and explores evolving new technologies ranging from ctDNA to ev to digital wearable devices. The paper is comprehensive integrating clinical and lab based data and platforms in this field and summarises ongoing clinical trials.
the paper appears comprehensive but as 50% of the references are not in the reference list it is hard to judge this completely. In supplement 1 one study is cited with not available beside the data - can the authors clarify ?
the paper is well written and comprehensive - it would be helped by more figures to make it less text dense - i would suggest a figure outlining the number of platforms used for her 2 assessment at present and ones under development; and i feel a figure similar to what the authors outline they have done for reference 118 in the paper to pull the themes mentioned together.
Round 2
Reviewer 1 Report
Comments and Suggestions for Authors
The authors have addressed my requested revisions with substantial additions that significantly improve the manuscript’s clarity and structure. The expanded sections on hormone receptor associations and molecular features are well developed and supported by appropriate citations.
Reviewer 3 Report
Comments and Suggestions for Authors
my comments in the prior review have been addressed